# COVID-19 and Diarylamidines: The Parasitic Connection

**DOI:** 10.3390/ijms24076583

**Published:** 2023-04-01

**Authors:** John Hulme

**Affiliations:** Department of Bionano Technology, Gachon Bionano Research Institute, Gachon University, 1342 Sungnam-daero, Sujung-gu, Seongnam-si 461-701, Republic of Korea; johnhulme21@gmail.com

**Keywords:** long COVID, serine protease, diarylamidines

## Abstract

As emerging severe acute respiratory syndrome coronavirus 2 (SARS-CoV-2) variants (Omicron) continue to outpace and negate combinatorial vaccines and monoclonal antibody therapies targeting the spike protein (S) receptor binding domain (RBD), the appetite for developing similar COVID-19 treatments has significantly diminished, with the attention of the scientific community switching to long COVID treatments. However, treatments that reduce the risk of “post-COVID-19 syndrome” and associated sequelae remain in their infancy, particularly as no established criteria for diagnosis currently exist. Thus, alternative therapies that reduce infection and prevent the broad range of symptoms associated with ‘post-COVID-19 syndrome’ require investigation. This review begins with an overview of the parasitic–diarylamidine connection, followed by the renin-angiotensin system (RAS) and associated angiotensin-converting enzyme 2 (ACE2) and transmembrane serine protease 2 (TMPRSSR2) involved in SARS-CoV-2 infection. Subsequently, the ability of diarylamidines to inhibit S-protein binding and various membrane serine proteases associated with SARS-CoV-2 and parasitic infections are discussed. Finally, the roles of diarylamidines (primarily DIZE) in vaccine efficacy, epigenetics, and the potential amelioration of long COVID sequelae are highlighted.

## 1. Introduction

As of 8 March 2023 (accessed: https://coronavirus.jhu.edu/map.html), more than 676 million COVID-19 cases and 6.8 million deaths had been reported globally. During that time, SARS-CoV-2 underwent significant mutations [1], resulting in Alpha (B.1.1.7), Beta (B.1.351), Delta (B.1.617.2), Gamma (P.1), and Omicron variants of concern (VOC). The most recent of these is the Omicron subvariant XBB.1.5, which is expected to be the aetiology of the majority of USA COVID-19 infections by early 2023 [2,3,4,5]. Unlike the original Wuhan strain, Omicron variants induce clinical symptoms associated with the upper respiratory tract and larynx, where the airway temperature is significantly lower [6], the viral replication rate is considerably higher [7], and the hosts’ immune response is reduced. Such conditions contribute to Omicron’s high transmissibility and ability to evade the hosts’ immune response.

The high transmissibility of Omicron has not only proved its infection and reinfection potential in humans and semi-domesticated species [8,9], but recent reports suggest the variant is prevalent in wild species [10]. Moreover, studies suggest that the requirement for P3 arginine in S1/S2 for furin cleavage common to different hosts provides a window for zoonotic transfer [11]. Accordingly, the World Health Organization has stressed the importance of veterinary and wastewater surveillance of *de novo* mutations and antigenic shifts in other species that could later appear in humans [12]. Water surveillance in cities and provinces has provided some unique insights into the emergence of the Wuhan and new Omicron strains and the transition from inter-pandemic to pandemic periods. In addition, previous studies have shown that increasing wastewater temperatures from 4 °C to 10 °C can more than double the decay rate of free-form SARS-CoV-2 RNA, measured by detecting the N1 and N2 genes [12]. However, water surveillance relates to viral identification and not the exact origins of the Wuhan and Omicron strains, the latter first being identified in South Africa on 24 November 2021 [13,14]. Consequently, this has resulted in several theories involving isolated pockets and viral reservoirs in immunosuppressed patients, rural communities, and mass migration of peoples (freelance contractors and refugees) [15] and animals (seasonal herds) being postulated regarding viral origins.

## 2. Bats, Pangolins, Parasites, and Diarylamidines: Making the Connection

An attempt to address the critical issue of the intermediate hosts of SARS-CoV-2 was recently presented by Tang et al. [16] in the general context of human coronaviruses but was left unresolved with additional vectors missing. Currently, the vectors associated with potential intermediate hosts (bats and pangolins) [17] include bat flies (*Diptera*: *Nycteribiidae* and *Streblidae*), bugs (*Hemiptera: Cimicidae* and *Polyctenidae*) that transmit blood ectoparasites (protozoans *Trypanosoma* spp. and *Plasmodium* spp.) [18,19], Wolkberg and Kaeng Khoi RNA viruses (*Bunyavirales*: *Bunyaviridae* and *Peribunyaviridae*), and dengue virus (*Flaviviridae*) [20]. A plethora of divergent viruses and different vectors found on live bats collected from Yunnan Province, China, between 2012 and 2015 was recently examined by Xu et al. [21]. Researchers encountered mainly arthropod-specific viruses and three possible arboviruses, suggesting that viral spillage between bats and common parasitic arthropods rarely occurs; whether this is the case for other parasite arthropods such as *Hemiptera* (triatomine bugs) could not be addressed. *Hemiptera* is renowned for its ability to transmit *Trypanosoma* spp., which are found in approximately 100 bat species worldwide [22]. The potential vectors for *T. cruzi* involve > 130 species of triatomine insects, five of which are epidemiologically more significant: *Triatoma infestans*, *Triatoma brasiliensis*, *Triatoma dimidiata*, *Rhodnius prolixus*, and *Panstrongylus megistus.* Currently, there are no documented cases of *Hemiptera* transmitting *Trypanosoma* species from bats to humans. However, just before the start of the pandemic, a related *Trypanosoma cruzi* genotype, specifically *T. cruzi* (TcBat), was reported for the first time in the brains of several bat species sourced from Yunnan Province (*Noctilio* spp., *Myotis* spp., and *Artibeus* spp.) [23]; unfortunately, the researchers did not test the blood or visit the animals to test for arthropods during the time of capture (2010–2015). TcBat was initially discovered in South America and has been detected in a Colombian child [24] and human mummies. Several bat families house the genotype, and phylogenic studies suggest that TcBat is a monophyletic lineage predominant in Colombia, Brazil, and Panama. In addition to TcBat, another bat trypanosome species reported to infect mammals and a handful of humans in Central America and north-western South America is *Trypanosoma rangeli* (*T. rangeli*) [25].

Before 2000, the origins of bat trypanosomes remained elusive until extensive evolutionary and taxonomic studies relating to the genus *Trypanosoma* resulted in the “bat-seeding hypothesis” proposed by Hamilton et al. in 2012. The hypothesis suggested that *T. cruzi* was the first bat trypanosome due to its expansive host range and employment of alternative vectors (triatomine bugs) permitting its transmission among New and Old-World terrestrial mammals. Additional multilocus phylogenetic analyses on multiple bat trypanosomal species sourced from Europe, South America, Africa, China, and Japan confirmed their placement within the *T. cruzi* clade, validating the hypothesis [25]. Hamilton further postulated that said clade probably arose from an ancestral group of isolated trypanosomes exclusively evolving in bats with multiple spill-over events into terrestrial mammals. The *T. cruzi* population is classified into six discrete typing units (DTUs): TcI to TcVI, with the recently named TcBat as the seventh. Trypanosomes within the *T. cruzi* clade are distributed into three main phylogenetic lineages, as shown in Figure 1.

*Trypanosoma* spp. are extracellular and intravascular blood parasites that cause debilitating acute or chronic disease in camels, cattle, humans, and other domestic animals. There are two phases of infection, acute and chronic, with the former associated with bone marrow hypoplasia anaemia, thrombocytopenia, and leukopenia in mammalian hosts [22]. Cardiomyopathy is the most common clinical outcome, and, until 2019, *Trypanosoma* infections and the associated disease (*Trypanosomiasis*) were endemic in 36 sub-Saharan African countries [26]. Strategies employed by trypanosomes to evade the host immune system include antigenic variation, induction of suppressor macrophages, myeloid-derived suppressor cells (MDSCs), and regulatory T cells [27].

Another parasitic disease [27] endemic on many continents [28] is babesiosis, a tickborne infection transmitted to humans by the bite of *Ixodes scapularis*, harbouring the parasite *Babesia microti (B. microti)* [29]. Prevalent in vertebrates, *B. microti* was recently found in 53 confiscated rare Sunda pangolins native to Thailand [27]. Moreover, additional reports found evidence of SARS-CoV-2-related coronaviruses circulating in Sunda pangolins [28]. Currently, there are no studies investigating *B. microti* and SARS-CoV-2 co-infections in humans or pangolins and the transfer of immunosuppressive factors, primarily due to the lack of testing and suitably available animal models. Furthermore, other reports show that co-infections involving *Babesia.* spp. [30] and *Trypanosoma.* spp. [31] are common with other parasitic species. In addition to blood parasites, several bacteria, including *Borrelia burgdorferi sensu lato* (*B. burgdorferi*)*,* benefit from co-infection with *Babesia*, which can prolong severe Lyme arthritis in mice when compared to controls [31,32].

However, exceptions do occur; for example, *Plasmodium* spp. infection prior to SARS-CoV-2 infection was reported to contribute to the disease’s poor prognosis and was seen to reduce the incidence of malaria in endemic regions [33,34,35]. Furthermore, additional investigations suggested that high exposure to malaria compared to low exposure prior to SARS-CoV-2 infection offered better protection against severe or critical COVID-19 [36].

Diarylamidines, diminazene aceturate (DIZE), and pentamidine have been used to treat *Trypanosomiasis* (*T. cruzi and T. brucei*—Chagas disease and African Sleeping. sickness), Leishmania, and Babesiosis in humans and cattle for decades [37,38,39,40]. DIZE and pentamidine are transported into trypanosomes, exerting their protozoalcidal abilities by targeting the adenine–thymine (A-T)-rich hairpin loops in circular and kinetoplast DNA resulting in the complete and irreparable nucleic acid loss. In addition, diarylamidines can also bind to mitochondrial II topoisomerase and transfer RNA [41]. Outside of their interactions with parasites, numerous studies have demonstrated that DIZE and pentamidine can act as bacterial growth inhibitors and adjuvants to common antibiotics [42,43,44,45,46,47,48,49]. Moreover, DIZE was shown to inhibit the production of lipopolysaccharide (LPS) endotoxins [50,51], whilst pentamidine interacts with LPS via lipid A [52,53] (Table 1). 

The modulation of LPS activity via arylamidines and the potential implications regarding potential treatments of numerous diseases associated with aberrant immune activation and coagulation have led to a flurry of animal model studies, including (i) amelioration of neurological pathologies (Alzheimer’s) [55]; (ii) cardiac (atherosclerotic plaque stabilization, cardiac fibrosis) [56,57,58]; (iii) pulmonary (hypertension) [59]; (iv) liver (injury and biliary fibrosis) [60]; (v) diabetic (type 1) [61]; (vi) acute radiation exposure/damaged tissue (ARE) [62] (Table 2).

### 2.1. Revisiting the Renin-Angiotensin System (RAS)

The RAS consists of two axes (classical and non-classical pathways), which regulate blood pressure, electrolyte balance, and vascular tone in numerous tissues, including the heart, kidney, brain, and endothelium. The classical pathway is initiated with the cleavage of ten peptides from angiotensinogen by renin, generating angiotensinogen I (Ang I). Ang I is then cleaved by the dipeptidyl carboxypeptidase angiotensin-converting enzyme (ACE) into the octapeptide, angiotensin II (Ang II). Ang II binds strongly to the angiotensin type 1 receptor (AT1R), inducing vasoconstriction, and weakly to the angiotensin type 2 (AT2R) receptor, the latter being associated with preventing post-ischaemic cardiac remodelling in mice [57]. In addition, AT2R signalling has been associated with the attenuation of lipopolysaccharide (LPS)-induced TNF-α and IL-6 production and increased production of anti-inflammatory IL-10 [50]. The remaining octapeptides can then be cleaved of an Asp group by aminopeptidase A resulting in the heptapeptide angiotensin III (Ang III) [64]. The heptapeptide is then metabolized by aminopeptidase N, resulting in Ang IV. Ang IV has been shown to restore short-term memory, spatial learning, and memory in double transgenic amyloid precursor protein (APP) mice [65]. Moreover, Ang IV has demonstrated potent antioxidant and neurogenic benefits independent of amyloidosis.

In the non-classical pathway, angiotensin-converting enzyme 2 (ACE2) converts the toxic octapeptide angiotensin II (Ang II) to the heptapeptides angiotensin 1–7 and alamandine. Furthermore, ACE2 also cleaves Ang I, generating angiotensin 1–9, which has been shown to protect the myocardium against ischaemia [63,66]. In addition, angiotensin 1–7 and alamandine can stimulate the Mas-related G-coupled protein receptor member (MRGDR) and Mas1 oncogene receptor (MasR) in blood vessels, increasing blood vessel relaxation and reducing mean arterial blood pressure (ABP). Furthermore, activation of this receptor reduces cardiac fibrosis in Sprague-Dawley rats by stimulating Src homology2-containing inositol phosphatase 1 (SHP-1), a redox-sensitive enzyme that impairs the signalling of molecules, such as mitogen-activated protein kinases (MAPKs), via dephosphorylation, thereby inhibiting the proliferative and profibrotic signalling induced by ANG II [67]. Moreover, angiotensin 1–7 (Ang 1–7) exerts an anti-hypertrophic effect by inhibiting the nuclear factor of activated T cells [67]. This anti-hypertrophic effect also depends on atrial natriuretic peptide (ANP) secretion during atrial pacing associated with activation of the Na+/H+ exchanger (NHE1) and calcium/calmodulin-dependent protein kinase II (CaMKII) via the phosphatidylinositol-3-kinase (PI3K)-protein kinase B (AKT) pathway [68]. In addition, the non-classical pathway can generate Ang (1–7) from endopeptidases neprilysin, prolyl endopeptidase, and thimet oligopeptidase [69]. Furthermore, ACE2 also hydrolyses a plethora of non-RAS-derived plasma-borne and inflammatory peptides, including [70] des-Arg9-bradykinin, neurotensin, dynorphin, and kinetensin.

As well as its modulating roles in RAS, ACE2 also counteracts low blood pressure in the kinin–kallikrein system (KKS). The KKS releases the active form of bradykinin (BK) from damaged tissues and converts to des-Arg-Bradykinin (DABK) and lys-Des-Arg-Bradykinin (BK1–9 and LysBK0–9). DABK is an active metabolite of BK that signals via the BKR1 receptor and is hydrolysed by ACE2 [70]. In addition, ACE degrades BK1–9 and LysBK0–9 in the KKS and is anti-inflammatory; in contrast to its role in RAS, accumulating active BK metabolites via the downregulation of ACE2 can worsen inflammatory reactions. The involvement of KKS remains controversial regarding severe COVID-19, although reports suggest [71] that it could play significant and minor hypotensive roles in critical and hospitalized patients.

### 2.2. SARS-CoV-2, ACE2, TMPRSS2, and Shredases

SARS-CoV-2, the beta coronavirus responsible for asymptomatic and symptomatic COVID-19 infection and reinfections adheres strongly to many cell lines primarily due to the high affinity of the unique S protein 13 amino acid region (13-mer) located within the RBD for ACE2 and the multitude of proteases that can readily cleave the S protein. The S protein of SARS-CoV-2 is only distinguishable from bat/Yunnan/RaTG13/2013 CoV (bat RaTG13) spike protein by four amino acids, PRRA at positions 681–684. The RBD is located at the tip of the N-terminal S1 subunit and, together with the C-terminal S2 subunit (responsible for fusion with the host cell membrane), constitute the active components of the homotrimer S protein responsible for infection [72]. Upon binding of S1 with ACE2, the ACE2–S protein complex pivots exposing the furin-cleaving site to type II transmembrane serine protease (TMPRSS2) on the host cell surface. TMPRSS2 is a transmembrane protein and cooperates with furin (catalytic serine protease domain of the subtilisin type) in the cleaving of ACE2 at arginine and lysine residues located at amino acids 697–716 [73], priming the C-terminal S2 subunit and subsequent fusion with host cell membrane, by-passing antiviral interferon-induced transmembrane proteins (IFITM) proteins. Other host proteases that participate in the priming of the S protein include cathepsin L, cathepsin B, trypsin, factor Xa, thrombin, and elastase [74]. As more ACE2 receptors are sequestered and the levels of endocytosed SARS-CoV-2 spike proteins and extracellular Ang II increase, the activity of additional shedding proteases, specifically ADAM17 (a disintegrin and metalloproteinase 17), is enhanced. This results in TNF-α (tumour necrosis factor-α) elevations in the extracellular environment, surface shedding of ACE2, and the initiation of a synergistic positive feedback loop that amplifies inflammation via active IL6 and IL1β [75,76]. A refresher of the SARS-CoV-2 infection cycle involving ACE2 and the intermediary proteins within the context of the RAS system is shown in Figure 2.

ACE2 expression was initially thought to account for mild and severe symptoms seen in most children and elderly during the early years of the COVID-19 pandemic. However, numerous reports have contradicted initial observations suggesting an inverse relationship between ACE2 expression and disease severity [77]. These contradictions were addressed in a study by Bastolla et al. [78], which showed that ACE2 mRNA and protein expression is not monotonically related to age. Moreover, the authors demonstrated that the expression of ADAM17 increases from birth, which predicts an increase in ACE2, implying that maximum expression would occur at an earlier age for serum protein [79]. Another factor that might account for the low risk of severe disease in children is the developmental regulation of expression of the TMPRSS2 gene, as reported by Schuler et al. [80].

Combination inhibitor therapies targeting TMPRSS2 and furin (aprotinin or MI-432 and MI-1851) have shown enhanced antiviral activity against SARS-CoV-2 in human airway cells compared with controls [81]. Furthermore, Yang et al. recently showed that bruceine A and gamabufotalin, when used together in the nanomolar range, could block the replication of the Delta and Omicron variants of SARS-CoV-2 [82]. However, the impact of small molecule inhibitors on the activity of factor Xa (catalytic triad), which has been shown to catalyse S1/S2 cleavage > an order of magnitude faster than TMPRSS2 [74], is yet to be reported.

The shredding of ACE2 from cell membranes by ADAM17 can occur for more than 35 days after infection, resulting in elevated serum ACE2 [83], and is associated with decreased activity of membrane-bound ACE2 and an increase in the active soluble form of ACE2 (sACE2) in plasma [83]. In addition, human recombinant soluble ACE2 (hrsACE2) has been proven to reduce SARS-CoV-2 recovery from Vero cells by a factor of 1000–5000 [84]. Combinatorial treatments utilizing extracellular vesicles (EVs) constituted of cellular ACE2 (cACE2) and TMPRSS2 were demonstrated to be much more effective than soluble ACE2 (sACE2) EVs for binding and retaining SARS-CoV-2 [85,86]. The increasing prominence of EVs, specifically hybrid membrane EVs and their potential application in targeted organ therapies, mRNA vaccination vehicles, and complex treatment algorithms are beyond the scope of this review.

### 2.3. ACE2 Potentiation and SARS-CoV-2

Direct DIZE potentiation of ACE2, encompassing enhanced receptor expression, mRNA levels, and substrate turnover (Ang II), remains a contentious issue. As discussed in the review by da Silva Oliveira et al. [87], the sources of contention can be found in the studies by Thatcher [88] and Velkoska al. [56], in which an acute hypertensive investigation [88] failed to demonstrate enhanced conversion of Ang II to Ang (1–7) in the presence of DIZE. In contrast, the Velkoska subtotal nephrectomy trauma rat study showed an increase in mRNA and ACE2 activity, with authors attributing said elevations to improved tissue injury and reduced ACE activation.

In an attempt to partially resolve the issue, Matsoukas et al. [64] postulated that the triazene proton of DIZE and the tetrazole proton of losartan possess a shared homology, permitting the former to play dual-purpose roles within RAS, that of putative angiotensin-converting enzyme 2 (ACE2) receptor activator and angiotensin type 1 receptor antagonist (AT1R). To support that hypothesis, the authors employed [64] a combination of in silico methodology and in vitro animal studies which showed that DIZE interacts with the spike protein, abolishing AngII-mediated constriction in rabbit iliac arteries akin to the commercially available AT1R inhibitor candesartan. Moreover, in silico structural modifications have shown that when DIZE is linked to remdesivir, the interaction of the antiviral with S-protein is significantly enhanced, thereby increasing its antiviral invasion potential [89].

Further, in silico modelling suggests that BAR107, BAR708, and ursodeoxycholic acid (UDCA) are ACE2 activators [90]. UDCA is a hydrophilic nontoxic bile acid that stimulates the expression of major histocompatibility complex antigens and is reported to reduce liver fibrosis in children [91,92]. However, recent work by Brevini et al. [93] showed that UDCA impaired farnesoid X receptor (FXR) signalling, downregulating ACE2 transcription and limiting SARS-CoV-2 infection in murine models. Moreover, human lungs and livers perfused with UDCA were less susceptible to SARS-CoV-2 infection, while viral transmission was significantly reduced; furthermore, a correlation between UDCA treatment and positive clinical outcomes after SARS-CoV-2 infection was identified. However, as a point of caution, the authors stressed that FXR activation upregulates ACE2 expression and has various functionalities, including hormonal regulation, glucose homeostasis, NF-κB modulation, and minimizing inflammation and fibrosis in numerous tissues. Finally, the authors highlighted the complex interplay between UDCA and inflammation, suggesting that the bile acid supplement would best serve as a prophylactic. Other FXR modulators include pentamidine (agonist) administered in the aerosol form [94] and ivermectin (antagonist) topical cream.

DIZE and UDCA’s essential functions have been elucidated via bile acid-sensitive ion channel (BASIC) investigations [95]. BASICs are related to acid-sensing ion channels (ASIC) and epithelial Na+ channels (ENaC) and are highly expressed by gut hepatocytes and cholangiocytes [96]. The channels adopt an open state in the presence of negatively charged amphiphilic molecules such as UDCA and a blend of hyodeoxycholic acid (HDCA) and chenodeoxycholic acid (CDCA). Conversely, diarylamidines, specifically DIZE and the calcium ion, were reported to strongly inhibit (semi-closed state) BASIC at micromolar concentrations, reducing the bile transepithelial current by 50%, with the remaining current utilized by other transport mechanisms [97]. DIZE’s ability to close the channel involves a cytosolic n-terminal inhibitory amphiphilic α-helix domain. In addition, DIZE’s narrowing or closing of other ion channels may collectively present as a negative membrane curvature, acting as a potential antagonist to the fused protein (FP) intermediary of SARS-CoV-2 [98]. Whether DIZE influences the fusion, packaging, and budding of SARS-CoV-2 and other virions is yet to be reported.

## 3. Diarylamidines, Serine Proteases

The triazene bridge and the two amidine groups present in the molecular structure of DIZE and, to some extent, pentamidine (amidine groups) are thought to be responsible for their interactions with single-stranded, double-stranded, and supercoiled DNA. [99]. These trypsin/mesotrypsin [100] substrate mimics (Figure 3) are charged at neutral pH (poor oral bioavailability) and are part of a group of diarylamidines (diminazene, pentamidine, hydroxysitlbamidine, and 4’,6-diamidino-2-phenylindole, (DAPI)) which inhibit the activity of all ASIC subtypes, including ASIC1a [101], whose blockade has been shown to be beneficial in animal models of Huntington’s and Parkinson’s diseases [102,103], ischemic stroke [104], and multiple sclerosis [105,106].

Due to its nanomolar potency for subtype ASIC1 and its inability (apart from stroke) to cross the blood–brain barrier (free form), DIZE is often limited to usage as a standard drug for in vitro modelling of new ASIC inhibitors [107]. Like many ASIC drugs (hydroxysitlbamidine and pentamidine) [108], diminazene, as well as its metabolite para-amino benzamidine (pABA), are serine protease inhibitors [109]. More recently, a cell-based proteolytic assay investigating the repurposing potential of DIZE and the clinically approved ASIC drugs camostat, nafamostat, and gabexate [110,111,112] for the treatment of mild COVID-19 further demonstrated DIZE’s broad spectrum ability to inhibit the catalytic triad of TMPRSS2 (HIS296, ASP345, and SER44), as well as furin’s active sites (ASN192, LEU227, SER253, ASP258, ASP306, THR367, and SER36). In an attempt to further understand DIZE’s ability to inhibit both proteases, additional modelling studies conducted by the authors revealed that the drug adopted a similar binding pose to that of the furin inhibitor [m-guanidinomethyl-phenylacetyl-Arg-Val-Arg-(4-amidomethyl)-benzamidine] MI-52. Additionally, the drug interacted favorably with residues at catalytic and binding sites of TMPRSS2 as well. Collectively, the results suggest that further studies utilizing DIZE in conjunction with other protease inhibitors regarding viral entry in animals and humans may be warranted.

However, DIZE’s ability to reversibly inhibit other proteases does not stop with SARS-CoV-2, specifically regarding the catalytic triad domain (HIS, ASP, and SER) of the trypsin-like peptidase Oligopeptidase B (OPB) [113]. OPB is part of a family of serine prolyl oligopeptidases that contribute to the virulence of Leishmania, Leishmania major, *L. amazonensis*, trypanosomes, *T. cruzi*, *T. brucei*, *T. evansi* [114,115], and other parasitic diseases, whose OPB catalytic activity favours the carboxyl side of pairing basic amino acid residues. The arginine at the P1 position renders it more vulnerable to trypsin substrate mimics such as DIZE and, to a lesser extent, pentamidine [116]. OpdB has been shown to inactivate atrial natriuretic factor (ANF) in the bloodstream of *T. evansi*-infected rats [117] by cleaving the hormone at four sites and is resistant to host plasma peptidase inhibitors [118] such as α2-macroglobulin, cystatins, kininogen, antiplasmin, and antithrombin III. Furthermore, OpdB can hydrolyse adrenocorticotropic hormone (ACTH), glucagon, neurotensin, angiotensin I, and vasoactive intestinal polypeptide, resulting in patient hormonal imbalance.

In addition to OpdB, *T. brucei* also utilizes other prolyl oligopeptidases (POPs). One of them is POP Tc-80, an enzyme involved in degrading large protein substrates such as collagens, fibronectin, and small peptides [119], accelerating the parasitic entry of macrophages and red blood cells (RBCs). Akin to OpdB, POP Tc-80 houses a catalytic triad, providing an additional target for DIZE and pentamidine. Moreover, the blood fluke *Schistosoma mansoni* responsible for *S. mansoni* infection (Schistosomiasis) also employs a prolyl oligopeptidase (catalytic triad Ser556, Asp643, and His682), aptly named *S. mansoni* (SmPOP) to infect muscle tissue, gastrodermis, and the gut lumen in animals and humans [54]. Unsurprisingly, when DIZE was intraperitoneally (IP) administered as a treatment for blood fluke-infected mice, it reduced the worm burden by 87% compared with praziquantel’s 92%, as well as serum alanine and aspartate aminotransferase (ALT) levels (liver markers) [120]. SmPOP is interesting not only because it plays an important role in the second most prevalent parasitic disease (*Schistosomiasis*) in sub-Saharan Africa and parts of Brazil [121], but because of its ability to cleave multiple RAS human peptides (Ang I and Bradykinin), which potentially allows it to modulate the RAS system. The ability of schistosomes to further influence the host is also reflected in work by Wang et al. [122], which demonstrated that the parasite could cleave the host co-factor kininogen, a regulatory enzyme involved in multiple stages of the coagulation and inflammatory processes within the host.

### 3.1. Immunosuppression, Vaccinations, and Epigenetic Potential

Protozoan immunosuppression can attenuate vaccine efficacy, limit immunological memory development, and reduce protection against co-infecting pathogens. Whether immunosuppression resulting from multiple co-infections has contributed to the emergence of SARS-CoV-2 and Omicron variants is open to debate. A recent review by Kamiya et al. [123] utilising an adaptive dynamics framework approach to epidemiological models with co-infection highlighted the evolutionary roles of immunosuppression, specifically the benefits to virulence and impairment to recovery (length of infection). Current indicators of reduced virulence and infection length associated with Omicron variants indicate the success of early vaccine programs. However, the risk of breakthrough infections and waning immunity following vaccination are increasing, particularly for those receiving B cell-depleting therapies for rheumatoid arthritis [123]. In addition, a recent report showed that a 4th booster shot resulted in significant IgG4 subtype elevations [124], consistent with immunosuppression, potentially predisposing individuals to other diseases. Host immunosuppression (T-cell exhaustion and chronic disease) resulting from protozoa or multiple SARS-CoV-2 variant infections is now well documented [125,126,127]. Thus, alternative treatments such as localized mucosal and oral-based vaccines [128], antivirals [129], and broad-based antiparasitics or combinations thereof are being considered.

Due to its high therapeutic index, DIZE is widely employed in treating trypanosomiasis in various livestock [130]. A concerning feature of trypanosomiasis, specifically *T. brucei* infection, is its ability to readily infiltrate the bone marrow and spleen, abolishing homeostasis, depleting a wide range of B cell populations in animal hosts, and reducing vaccine efficacy [131]. For example, an investigation [132] by De Trez et al. (2015) showed that *T. brucei* could block the development of B cell-mediated collagen-induced arthritis in DBA-prone mice and the survival of anti-type collagen antibodies. Interestingly, DIZE treatment restored the clinical signs of the disease, with authors attributing the reversal to the restoration of specific autoimmune antibody levels. However, a recent study by the same group showed that DIZE treatment did not restore the thymo-dependent and independent humoral response or the vaccine-induced memory response (antibody levels) in malaria-vaccinated C57BL/6 mice following *T. brucei* infection [133]. Moreover, the authors proposed that vaccine memory was partially destroyed or the accessibility to those memories was prohibited by infection, further highlighting the need for malaria vaccine trials in areas where trypanosomiasis is prevalent. The influence of protozoan infection on bacterial and viral vaccination efficacy and the adaptive immune response is summarized in Figure 4 [34].

Other pathogens capable of interfering with the development of efficient humoral immune responses include *Salmonella* bacteria, known to reduce the number of IgG-secreting plasma cells in bone marrow [134], and the highly infectious measles virus (MeV), which can deplete previously expanded B memory clones [135], leading to a loss in vaccine acquired immunity. Another factor that might account for the loss of vaccine efficacy and suppression of the immune response is extracellular vesicles (EVs). Interestingly, the ability of the host to produce EVs and initiate a strong inflammatory response could rest in the activity of the ATP-binding cassette transporter A1 (ABCA1). ABCA1 knockout mice exhibit a reduced capacity to produce EVs, which are thought to reduce the severity of cerebral malaria in patients. Conversely, increased circulating EVs sourced from platelets, erythrocytes, and endothelial cells correlate with a high fever. The immunomodulatory interplay between host and parasitic EVs was recently addressed by Wu et al. [136]. For those readers interested in the application of hybrid vaccination EVs generated from parasitic and host cell membranes, the review by Khosravi et al. is recommended [137].

A parasite or pathogen can also influence the host’s immune response via various epigenetic mechanisms, perturbing the regulation of expressed and non-expressed genes, resulting in homeostatic imbalance. For example, several pathogenic infections, Mycobacterium tuberculosis (TB) [138], Schistosomiasis [139], Hepatitis C (HCV) [140], lymphocytic choriomeningitis virus (LCMV) [141], and sepsis [142] engage in immune suppression by employing a plethora of epigenetic mechanisms, ranging from the acetylation (permissive access) and methylation (restricted access) of histones H3 [138] and H4 to the hypo and hypermethylation of interleukin-4 (IL-4) and Th1 pathways [139], evoking a polarized immune response [143]. There are four core histones, H2A, H2B, H3, and H4, with H3 constituting the major site of histone methylation. Under the catalysis of histone methyltransferase (HMT), a methyl group is transferred from S-adenosine methionine to the histone lysine residues. Conversely, histone demethylases (HDM) regulate the removal of methyl. In sepsis, multiple methyltransferases induce changes in inflammatory signalling and related inflammatory factor expression by catalysing histone methylation [139]. Histone modifications differentially regulate the genes of macrophages, permitting them to alternate between the classically activated (IFN-γ or LPS) M1 and the three types (M2a, M2b, and M2c) of alternatively activated (IL-4, IL-10, or IL-13) M2. The activation of M1 and M2 macrophages is associated with sepsis’s pro and anti-inflammatory stages.

Work by Stachowicz et al. showed that DIZE treatment increased anti-inflammatory M2 macrophages in attenuated hepatic steatosis in apoE-knockout mice [59]. Moreover, the authors reported decreased levels of triglycerides and elevated plasma levels of HDL and taurine related to the liver, which may indicate ABCA1 involvement. Taurine is one of the most prevalent amino acids in muscle tissues (skeletal and cardiac), with transporters found in proximal cells of the kidney, neuronal plasma membranes, and endothelial cells in the blood–brain barrier, rendering it critical to neuroendocrine function [144]. In addition, taurine alleviates repression of betaine-homocysteine S-methyltransferase (BHMT), a zinc-dependent methyltransferase [145,146] that uses betaine (methyl donor) in the methylation of homocysteine to methionine, which, in turn, supports S-adenosylmethionine (SAM) biosynthesis in humans.

SAM is a potent anti-inflammatory that reduces LPS-induced TNF−α expression, inhibits IL-6-dependent signalling, and modulates the polyamine recycling pathway [88]. A key enzyme in the pathway is spermine/spermidine N1-acetyltransferase (SSAT1), which DIZE inhibits [147]. Previous research combining SAM with DIZE in the treatment of isolated synovial fibroblasts sourced from patients with rheumatoid arthritis (RA) or osteoarthritis (OA) resulted in elevations of 5-methylcytosine (5-MeC) and DNA methyltransferase 1 (DNMT-1), and reduced the adhesion and invasiveness of RA synovial fibroblasts (RASFs) in a mouse model.

In addition to DIZE, pentamidine also demonstrates epigenetic potential when combined with mitomycin C and ionizing radiotherapy, reducing histone H2A acetylation, tumour growth, and DNA double-strand breaks (DSBs) in HeLa cells [148]. Moreover, the recently mined lysine-specific histone demethylase 1 (LSD1) developmental inhibitor 304 structure [149] is also closely analogous to DIZE and pentamidine.

### 3.2. Long COVID and DIZE

It is well accepted that upon SARS-CoV-2 infection, the virus can induce a state of immunosuppression and hormonal imbalance in the host (akin to a parasite) by hijacking the epigenetic mechanisms associated with lymphopoiesis control (CD4+ and CD8+), the degree to which depends on disease severity. If the infection is severe, there can be resultant hypermethylation of IFN and immunosuppressive genes, leading to dysfunction. Numerous studies have linked immune dysfunction to the development of long COVID syndrome [150,151,152]. Patients reporting long COVID exhibit a variety of symptoms involving multiple organ systems. Studies reveal that 90% of survivors develop chronic fatigue and numerous respiratory, neurological, cardiac, and renal manifestations [153]. In 2022, the Delphi consensus broadly defined long COVID as the presentation of symptoms (3 months post-infection) with varying time courses (fluctuating, increasing, relapsing, new onset, persistent) lasting a minimum of 2 months [154]. One of the most important predictors for the underlying pathologies observed in ‘Long COVID’ is endothelial dysfunction and thrombosis, resulting in misfolded fibrin (amyloid-like microclots), resistant to fibrinolysis [155]. Kell and Pretorius et al. [156] recently demonstrated that extensive fibrin amyloid microclots persist in long COVID patient samples. Moreover, a small one-month clinical study [157] involving a subset of 24 patients treated with antiplatelet therapy (DAPT) (Clopidogrel 75 mg/Aspirin 75 mg) once a day and a direct oral anticoagulant (DOAC) (Apixiban) of 5 mg twice a day showed reductions in fibrin amyloid microclots and blood platelet pathology scores, resulting in sufficient endothelium recovery and the resolution of primary symptoms.

Whether amyloid microclots could explain the elevated amyloid beta-42(Aβ-42) and amyloid beta-40 (Aβ 40) serum levels reported in chronic obstructive pulmonary disease patients [158,159] and the link with mild cognitive decline (MCI) is debatable. Other attempts to address some of the multiple pathogenic mechanisms (multiple organ dysfunction (MOD) and acute respiratory distress syndrome (ARDS)) also involve aspirin and dapsone [160]; the latter considered a second-line treatment of immune thrombocytopenia [161] and linked to the regulation of mild cognitive impairment (MCI) in Alzheimer’s disease (AD) [162].

In addition to dapsone, another DNA minor grove binder that alleviates AD pathology, exhibiting an idiosyncratic drug reaction in animals, is DIZE [163]. Unfortunately, its toxicity prevents its application in a clinical trial; thus, a model/models that encompass severe COVID-19 pathologies (posing a higher risk of long COVID), such as those observed in hematologic acute radiation syndrome (H-ARS) and multiorgan delayed effects of acute radiation exposure (DEARE), in rats might suffice. Recent work by Gasperetti [64] utilizing an H-ARS rat model showed that IV-administered DIZE treatment increased 30-day survival by 30% compared to vehicle control rats following an LD50/30 total-body irradiation (TBI) dose of 7.75 Gy. In addition, the authors investigated whether long-term administration of DIZE could mitigate the major sequelae of radiation injury involving the bone marrow, gastrointestinal tract, lungs, and kidneys. It was found that treatment increased median survival from 147 days in control-irradiated rats to 177 days. The authors attributed the extension to a unique signalling mechanism, including the increased transcription of RAS receptors AT2R and MasR counterbalancing AT1R receptor function, promoting vasodilation, and reducing inflammation and fibrosis. In conclusion, DIZE administration after radiation injury promoted multiorgan recovery without overt tissue toxicity. Whether DIZE could alleviate some of the associated sequelae observed in long COVID-18 [164] (postural orthostatic tachycardia syndrome, neurological dysfunction signalling, immune dysregulation, microbiota disruption, and erectile dysfunction [165,166]) in animals or humans is yet to be proposed.

## 4. Conclusions

In 2015, da Silva Oliveira and de Freitas [88] drew attention to the diarylamidine DIZE and its interaction with RAS, implying that the compound could modulate disease severity by prioritizing the non-classical pathway via ACE2 activation, whilst offsetting the deleterious effects of Ang II. At that time, many of the diarylamidines were also recognized as broad-spectrum serine protease inhibitors in parasitic and animal model investigations, suggesting that they could play a role in stabilizing uncontrolled proteolysis (septic shock) when used with other drugs. This, in turn, led to pentamidine and DIZE being used synergistically in numerous drug candidate investigations involving a plethora of diseases, including mental illnesses, Huntington’s disease, hypertension, myotonic dystrophy, autoimmunity, diabetes, cancer, and, of course, parasitic infection (hydroxychloroquine, chloroquine, and azithromycin), resulting in several patents [88]. In 2018, additional investigations highlighted the potential role of DIZE in preserving vaccine efficacy and rebalancing the methylation status in the host (epigenetics). The following year brought the COVID-19 pandemic, and the central role of RAS in SARS-CoV-2 infection was identified; thus, the potential of diarylamidines to inhibit the action of several proteins involved in the infection cycle was naturally researched.

This review examined the results of numerous *in vivo* and *in vitro* investigations utilizing diarylamidines (mainly DIZE) and their inhibitory roles in SARS-CoV-2 infection. In line with expectations, diarylamidines exhibited broad spectrum potential by activating ACE2 and inhibiting various intermediary proteins critical to the viral infection cycle. In addition, it raised the possibility that DIZE, in conjunction with other compounds, could be used to minimize and ameliorate symptoms associated with severe and long COVID, such as organ damage, erectile dysfunction, amyloid clots, excess fibrosis, and immunological dysfunction.

The downside of this review is that it does not address the toxicity issues surrounding the administration of free-form diarylamidines; although, in retrospect, this was addressed more than a decade ago [87] with the advent of micro- (slow-release liposomal formulations and thermo hydrogel preparations), and nano-encapsulation-targeted therapies [167,168].

Suggestions for further laboratory research might include DIZE’s potential role in the immunomodulatory interplay between host and pathogenic EVs (parasitic and microbial) and its ability to modulate the vesicle content and blebbing processes. In addition, its synergistic properties could be explored with a known antiviral and biodegradable nanocarrier such as chitin. Finally, regarding animal models and possible clinical trials, future investigations might include DIZE’s ability to slow the progression of vascular dementia by enhancing O_2_ carrying capacity, possibly in combination with exercise or moderate hypoxic conditioning [169].

## Figures and Tables

**Figure 1 ijms-24-06583-f001:**
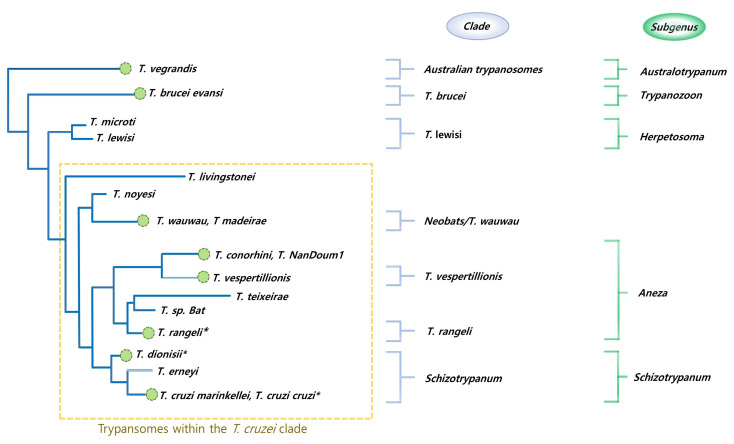
The phylogenetic relationships shared by bat trypanosomes and corresponding clade and subgenera nomenclature are represented. Zoonotic bat trypanosomes have an asterisk. Modified with permission [22].

**Figure 2 ijms-24-06583-f002:**
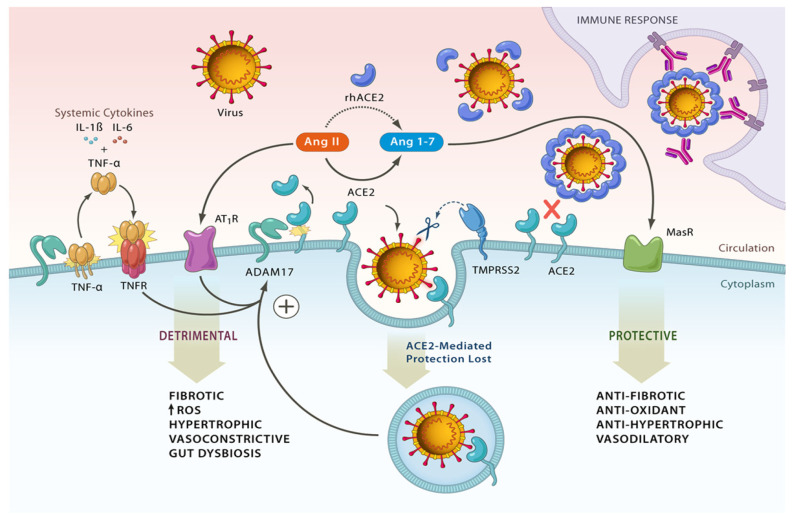
SARS-CoV-2 infection cycle involving ACE2 and the intermediary proteins within the broader context of the RAS system. Endocytosis of the ACE2 enzyme and severe acute respiratory syndrome-coronavirus (SARS-CoV-2) viral particles initiate the cycle. Following on, Ang II (angiotensin II) levels rise with increased activity of angiotensin 1 receptors (AT1R), fostering elevated ROS, vasoconstriction, and hypertrophy, isolating the Ang 1–7-driven branch of the RAS system. This activates the surface shredding enzyme (shredase) ADAM17, which cleaves ACE2 and is upregulated by endocytosed SARS-CoV-2 spike proteins. With upregulation, the enzyme cleaves its primary substrate, releasing soluble TNF-α (tumour necrosis factor-α) into the extracellular region with systemic cytokine elevations. Reproduced with permission from the American Heart Association [76].

**Figure 3 ijms-24-06583-f003:**
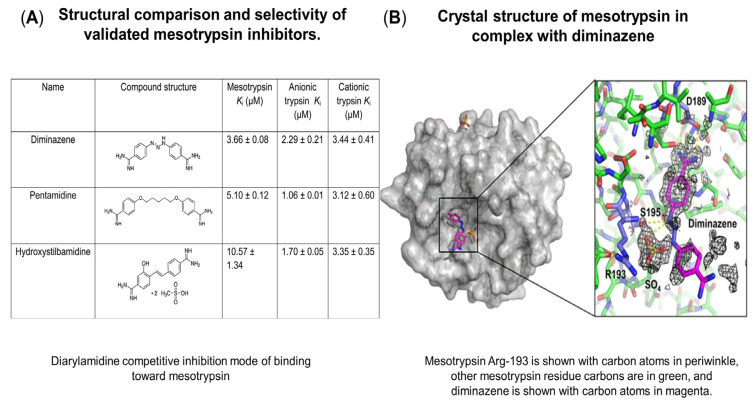
Small molecule inhibitors of mesotrypsin. (**A**) Competitive inhibition assay, all three compounds bind to mesotrypsin, with diminazene demonstrating the strongest affinity for the isoform. (**B**) Crystal structure of the diminazene mesotrypsin complex, interaction and specificity are mediated by Asp-189. Reproduced with permission [100] Creative Commons Attribution License.

**Figure 4 ijms-24-06583-f004:**
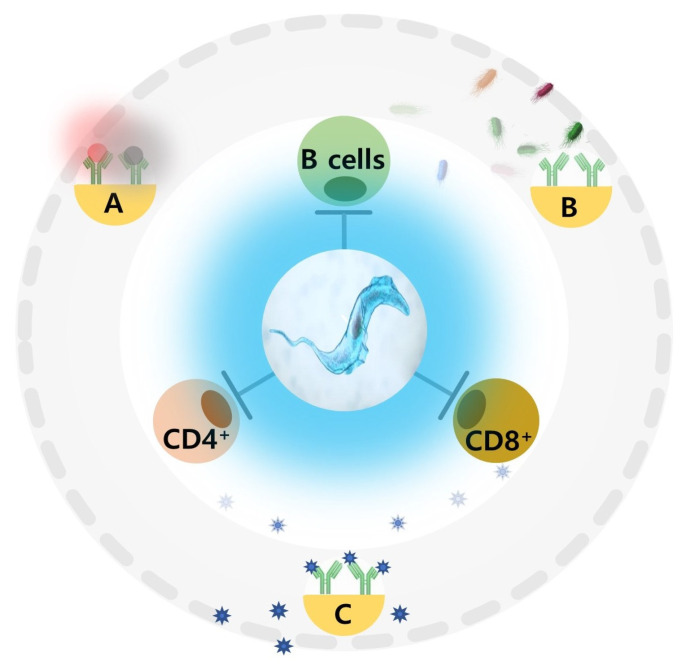
Parasitic impairment of B and T cells results in suppression of the adaptive response and specific antibody levels. (A) Decrease in specific antibody levels, increase in bacterial burden, exacerbation of disease severity, and the emergence of resistance to antimicrobials. (B) Diminished cellular humoral immunity and vaccine efficacy, possible breakthrough infection. (C) Protozoan modulated vaccine antibody response against different viral pathogens; impact on breakthrough infections inconclusive. Adapted from Akoolo et al. [34].

**Table 1 ijms-24-06583-t001:** Antiparasitic and synergistic capabilities of diminazene aceturate and pentamidine when used separately or in conjunction with antibiotics.

Diarylamidines & Combinations	Pathogen/Disease	Hosts/Experimental	Findings	References
DIZE & AZI	*T. brucei*	Albino Rats	↓ Parasitaemia ↓ Leukocytosis	[45]
DIZE	*T. congolense*	BALB/c mice C57BL/6 mice	↓ LPS-induced production ↓ IL-6, IL-12, TNF and IFN-γ	[42]
DIZE	*S. mansoniex vivo*	mice	↓ Aminotransferase levels ↓ Immature and adult parasites	[54]
DIZE	*T. congolense*	Pre-treated macrophages	↓ Phosphorylation of mitogen-activated protein kinase (p38), STAT1 and STAT3	[50]
CLD, DIZE, ACE, IMD	*C. babesiosis*	Dogs	↓ Malaise, fatigue, chills, fever	[40]
DIZE, CHL, STP	Antibiotic-resistant *K. pneumoniae*, *S. aureus* and *E. coli*	Minimal inhibitory concentrations (MICs) assay	↑ Gram-negative and positive bacterial sensitivity to antibiotics ↓ in MIC	[44]
PENT, colistin	*T. brucei rhodesiense*	66-year-old female	↓ Parasitaemia by 75% (trypomastigotes)	[39]
PENT	*A. baumannii*	Murine Infection Model	↑ sensitivity of colistin-resistant *A. baumannii* to gram-positive antibiotics	[47]
PENT analogue (P35)	*A. baumannii* & *K. pneumoniae*.	Murine model	↑ sensitivity of colistin-resistant *A. baumannii* to gram-positive antibiotics	[48]

Diminazene aceturate (DIZE), Pentamidine (PENT), Chloramphenicol (CHL), Streptomycin (STP), Imidocarb dipropionate (IMD), Clindamycin (CLD), Azithromycin (AZI), Aceturate (ACE). *Trypanosoma brucei* (*T. brucei*), *Trypanosoma congolense* (*T. congolense*), *Klebsiella pneumoniae* (*K. pneumoniae*), *Schistosoma mansoniex vivo* (*S. mansoniex vivo*), *Canine babesiosis* (*C. babesiosis*), *Acinetobacter baumannii* (*A. baumannii*), *Staphylococcus aureus* (*S. aureus*), *Escherichia coli* (*E. coli*).

**Table 2 ijms-24-06583-t002:** Amelioration of cardiac, neurological, kidney, and systemic experimental models by diminazene aceturate (DIZE).

Diarylamidine	Disease/Pathology	Hosts/Experimental	Findings	References
DIZE	AD	SAMP8 mice	↓ Neuropathology	[55]
DIZE	Liver injury & BF	MDR gene-2 knockout mice	↓ NOX enzyme assembly and ROS generation ↑ myofibroblasts & tissue repair	[60]
DIZE	ATP & HPS	ApoE-Knockout mice	Modulating macrophage response & taurine biosynthesis	[58]
DIZE	CF & DAD	W rats	↑ Protective effect on the heart under the pathological condition of kidney injury	[56]
DIZE	PHY	SD male rats	↑ Vasoprotective axis of the LRAS, ↑ pulmonary vasoreactivity, ↑ enhanced cardiac function, ↓ inflammatory cytokines	[57]
DIZE	NPP	W diabetic male rats	↑ Glomerular ACE2 & AT2 receptor expression ↓ fibrosis and apoptosis	[61]
DIZE	MORI	WAG/RijCmcr rats	↑ Survivability in rat models of H-ARS and DEARE	[62]
DIZE	CAR	W rats	↑ Acute antiarrhythmic-mic potential *in vivo* modulation of cardiomyocytes contraction and excitability properties	[63]

Alzheimer’s disease (AD), Senescence Accelerated Mouse (SAMP8), Biliary fibrosis (BF), Multiple Drug Resistant (MDR), Atherosclerotic Plaques (ATP), Hepatic Steatosis (HPS), Cardiac Fibrosis (CF), Diastolic Dysfunction (DAD), Pulmonary Hypertension (PHY), Nephropathy (NPP), Multiorgan Radiation Injury (MORI), Ischemia (IC), Myocardial Infarction (MI), Wistar Rats (W rats), Sprague-Dawley Rats (SD rats), Cardiac Arrhythmia (CAR), Hematologic Acute Radiation Syndrome (H-ARS), Multiorgan Delayed Effects of Acute Radiation Exposure (DEARE), Lung Renin-Angiotensin System (LRAS).

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
