# Peer review of "COVID-19 and Diarylamidines: The Parasitic Connection"

_ijms, 2023, doi:10.3390/ijms24076583_

Round 1

Reviewer 1 Report

Dear author,

Congratulations! Diagnostic of Omicron need your Review, of importance in clinical practice.It will be interessting to disscuss, may be in the mextcpaper, a problem of telomere schortening in the Long Covid.

At the line 452 of your papet you have a technical mistake: (skeletal-, cardiac), but without "brain'.

Fitst paper of Pretorius et sl., is number 157 in References. You could see at References once more, to correct small mistakes.

Best regatds, Reviewer

Author Response

Thanks for the feedback

corrected as suggested

A new Figure  has been added and an adapted version of Figure 4

Reviewer 2 Report

The author has made good contribution toward recent issue. 

In line 327 change 'in vitro' to 'in vitro'

Add the structure of camostat, nafamostat, and gabexate

In line 394 T.brucei to T.brucei and also through out 

In line De Trez et al add year and also follow through out the text

Author Response

Thanks for the feedback

corrected as suggested

PS  a new Figure  has been added and an adapted version of Figure 4

Reviewer 3 Report

Hulme presents a review on the parasitic connection between COVID-19 and diarylamidines. Given that the COVID-19 pandemic remains a global public health threat, it is essential to improve a broad spectrum insights on the disease. Thus the manuscript is timely; in addition, it is well written, except for a few minor grammatical and formatting issues.

Specific comments are as follows:

1.      Line 26: Please replace the “were” with “had been”.

2.      Line 29: Please replace the “constitute” with “be the etiology of”.

3.      Line 40: Please write “world health organization” as a proper noun.

4.      Line 41: Please italicize “de novo”.

5.      Lines 47 to 49: The sentence that begins with “However…” does not read well.

6.      Line 54: Please rewrite “host/s” as “host(s)”.

7.      Line 56: Please rewrite “unresolved and” as “unresolved, with”.

8.      Line 57: Please pluralize “Pangolin”.

9.      Line 101: The author begins the sentence with “To the best of our knowledge”, despite being the only author listed on the paper.

10.  Tables 1 and 2: The titles should be separated from the description of the abbreviations, with the said descriptions presented as footnotes to the tables.

11.  Line 138: Please rewrite “table 2” as “Table 2”.

Line 149: Please pluralize “pathway”.

Author Response

(The authors gave the same response as above.)
